# Intersection-Based Unicast Routing Using Ant Colony Optimization in Software-Defined Vehicular Networks

Hao Zhu, Jingru Liu, Li Jin and Guoan Zhang *

School of Information Science and Technology, Nantong University, Nantong 226019, China
* Correspondence: gzhang@ntu.edu.cn

**Abstract:** A vehicular ad hoc network (VANET) is a mobile ad hoc network composed of communication between vehicles, between vehicles and roadside units, and between vehicles and pedestrians, in order to achieve traffic safety and entertainment services. The design of the routing protocol is very important for the realization of the service function of VANET. Local optimum and network congestion problems are restraints of traditional geographic routing protocols for VANET. In this paper, a software-defined network (SDN) based unicast routing scheme in an urban traffic environment is proposed, which uses Dijkstra's algorithm to find a global optimal anchor path. The RSU neighbor discovery protocol is proposed, through which each RSU can discover its neighbor RSUs, and then each RSU periodically sends ant packets to its neighbor RSUs, evaluates the communication connection quality of each street segment according to the statistical data of the ant packets received, and sends the evaluation value to the SDN server in time. The SDN server has the connection quality evaluation values of all street segments in the global scope, from which an optimal anchor path can be calculated. The simulation results show that the proposed scheme has better packet delivery ratio than other related schemes.

**Keywords:** software-defined network; routing protocol; vehicular ad hoc network; ant colony optimization

## 1. Introduction

The main goal of vehicular ad hoc network (VANET) is to provide safety-related services for vehicles. Vehicles on the road periodically exchange information such as location, speed and so on, so that vehicles can perceive the surrounding traffic state, effectively improving traffic safety and travel efficiency. VANET can also provide infotainment services to make passenger journeys more enjoyable. Vehicular ad hoc networks have some different characteristics from other mobile ad hoc networks, such as the high mobility of vehicles and frequent changes of topology, which pose challenges to the design of packet routing protocols [1–5].

Topology-based routing protocols and position-based routing protocols are common routing protocols in VANET [6]. Topology-based routing protocols [7] use the network topology to design routing protocols, in which each node needs to adjust and maintain its own routing table according to the dynamic changes of the network topology. Topology-based routing protocols can be divided into proactive routing protocols and reactive routing protocols according to the different ways of establishing routing tables. Proactive routing protocols, such as DSDV [8] and OLSR [9], are also called table-driven routing protocols, in which each node uses a periodic route broadcast to exchange routing information and maintain its routing table [10]. The delay of this routing protocol is small, but the protocol needs a large number of broadcast route control messages, which has a large overhead. In addition, the dynamic change of vehicle speed and location leads to the instability of the routing table. Reactive routing protocols, also known as on-demand routing protocols,

such as AODV [11] and DSR [12], do not generate routes in advance, but establish routes when the source node has a message to send.

In the position-based routing protocol [13–15], each node is equipped with a Global Positioning System (GPS) or Beidou positioning system, which can easily obtain the position information of the node. Each node can achieve end-to-end routing to the target node through its own position, the position of the target node, and the position of the neighbor node. Each node does not need to periodically exchange information to maintain the routing table, so the position-based routing protocol has a lower network load compared with the topology-based routing protocol.

Position-based routing protocols often use a greedy mechanism to forward data packets, such as a GPSR protocol [16], and each node always tries to forward data packets to the nearest neighbor node from the target node. However, there are problems when used in urban environments, such as the impact of building obstacles on the signal, routing loops and wrong directions. In order to alleviate these problems, a geographical source routing (GSR) protocol [17] is proposed. In this protocol, the source node chooses Dijkstra's shortest path algorithm to calculate the shortest path to the destination node, but the protocol does not consider the network connectivity of vehicles in the road. For this reason, many connectivity-aware routing protocols have been proposed [6,10,17,18]. BETA [10] is a VANET routing protocol that uses periodical beacons among vehicles instead of control packets to realize traffic awareness, and reduces the network overhead. In [19], the vehicle density is used to evaluate the network connectivity quality of the street segment, and implicitly assumes that the vehicle density of the street segment is uniformly distributed. In [18], SDCR is a connectivity-aware routing protocol, which calculates the network connectivity of the street segment using both the vehicle number and vehicle distribution of the street segment, and improves the routing performance of VANET.

The network connectivity quality of a street segment is related to several factors, such as the density, speed and mobility model of vehicles. As for vehicle density, due to the randomness of driving behavior and the influence of traffic lights or traffic accidents, the vehicle density of the street segment is unevenly distributed, for example, as shown in the Figure 1, where case A may be caused by a red traffic light, traffic accidents can lead to case B, and case C is that the street segment is relatively smooth. Therefore, accurate modeling of the network connectivity quality of street segments is a challenging problem.

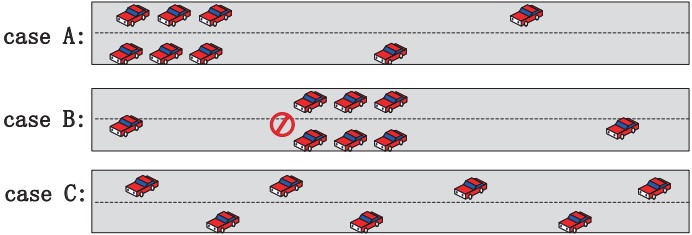

**Figure 1.** Distribution of different vehicle densities. Case A: traffic light influence, case B: traffic accident influence and case C: smooth street

Recently, bio-inspired approaches have been used for the routing protocol due to their ability to solve hard, np-hard and np-complete optimization problems [20,21]. Among bio-inspired optimization techniques, ant colony optimization (ACO) has received a great deal of interest [6,22]. These papers regard the network connectivity quality of the whole street as a black box, regardless of internal factors such as the density and distribution of vehicles, and uses the ant pheromone to evaluate the network connectivity quality of the street, so as to calculate the corresponding weights of network connectivity quality. However, most of these routing methods for VANET are distributed, which can not effectively use the global network information, resulting in local optimum and network congestion problems [23].

Lately, software-defined VANET (SDVN) has attracted much attention, which has the advantage of centralized control by combining software-defined network (SDN) ar-

chitecture with VANET [23–27]. SDN is an emerging programmable network architecture developed to simplify the architecture of traditional networks, and it uses a centralized control plane and a distributed forwarding plane, which are separated from each other. The control plane centrally controls the network devices on the forwarding plane and provides programmability, which greatly improves the flexibility and scalability of the network. In [23], the central controller maintains a routing table. Fuzzy logic and reinforcement learning are used to initialize and update the routing table, and then calculate the best route according to the link stability in the routing table. PT-GROUT [26] regards the vehicular network as a temporal graph, uses Hidden Markov Model (HMM) to predict temporal information, and calculates the optimal route path under the SDVN architecture.

This paper proposes a routing protocol that combines topology-based routing and position-based routing, where topology-based routing is used at the RSU level, and position-based routing works at the vehicle level. The ant colony optimization algorithm is introduced to calculate the optimal route path under the SDVN architecture. The main contributions of this paper can be summarized as follows. Firstly, a neighbor RSU discovery protocol is proposed, through which each RSU can find all its neighbor RSUs and prepare for sending ant packets between adjacent RSUs. Secondly, the ant colony algorithm is applied to two adjacent RSUs of a street segment, and ant packets are periodically sent between the adjacent RSUs to calculate the evaluation value of the connectivity quality of the street segment. Finally, SDN is applied to VANET, and the RSU sends the evaluation values of the relevant street segments to the SDN server, which holds the connectivity quality information of all street segments and calculates the optimal route in the global scope according to the routing request.

The remainder of this paper is organized as follows. In Section 2, the proposed routing scheme is presented, mainly including neighbor RSU discovery protocol and SDN-based routing protocol. The simulation tools, scenario, parameters, an evaluation of the results, and a comparison against the GSR and GPSR are described in Section 3. Finally, Section 4 concludes this paper.

## 2. Materials and Methods

In this section, a SDN-based routing scheme using ant colony optimization is presented, which can effectively improve the packet delivery rate. Additionally, a new neighbor RSU discovery protocol is proposed, which can find the neighbor RSUs for each RSU, and facilitates the transmission of ant packets between adjacent RSUs.

### 2.1. System Model

Assuming that RSUs are deployed at both ends of each street segment, as shown in the Figure 2, the RSUs are standalone, i.e., not directly connected to other RSUs by wire, but they can communicate indirectly through the base station. For the P2P multimedia data exchange between the vehicle nodes in VANET[28], the traffic cost through cellular communication alone is very expensive. Therefore, the data packets can be transmitted through the multi-hop routing between the vehicles, and some basic control information can be transmitted using cellular communication. When the vehicle node needs to send data, it only needs to attach the necessary control information to the header of the data packet and deliver it to the nearby RSU, which is responsible for forwarding packets. The communication between adjacent RSUs needs to forward packets through multi-hop routing through the vehicle nodes on the street segments. In addition, each vehicle is equipped with digital maps and on-board navigation systems, such as Beidou Navigation or GPS.

The architecture of a software-defined VANET in this paper is shown in Figure 3. The RSU can communicate with the SDN server in the simulation area in both directions. Each RSU periodically sends the connectivity quality evaluation values of its adjacent street segments to the SDN server, so that the SDN server holds the global information of all the street segments in the simulation area and can calculate an optimized route.

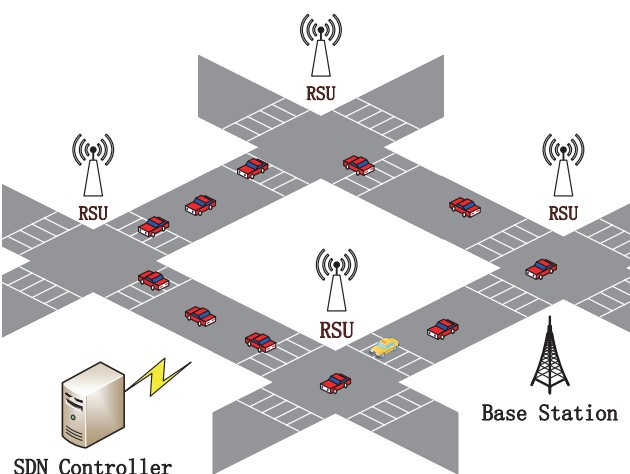

**Figure 2.** Structure diagram of the system.

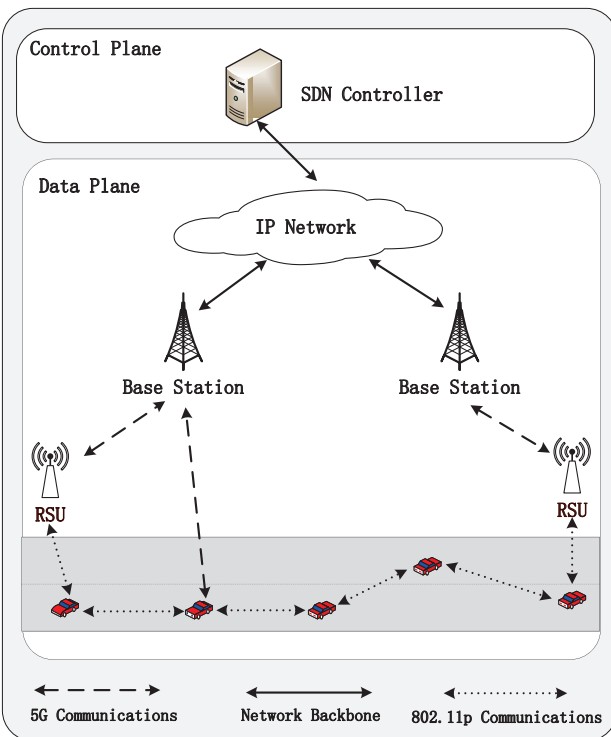

**Figure 3.** Software-defined VANET architecture.

### 2.2. Neighbor RSU Discovery

We take the communication delay between the adjacent RSUs at both ends of the street segment as the external characteristic to evaluate the connectivity quality of the street segment, and calculate the evaluation value of the connectivity quality through the ant colony algorithm. Based on the connectivity quality of each street segment, the Dijkstra shortest path algorithm is used to calculate the optimal path.

A neighbor RSU of one RSU is the RSU in the same street segment and it does not exist with any additional RSU in the middle. For example, in Figure 4, the neighbor RSUs of $RSU1$ are $RSU2$ and $RSU4$, which are pointed out by the dotted arrow; the neighbor RSUs of $RSU5$ include $RSU2$, $RSU4$, $RSU6$ and $RSU8$, which are indicated by the blue dashed arrows; and $RSU5$, $RSU7$, and $RSU9$, which are identified by red solid line arrows, are neighbor RSUs of $RSU8$.

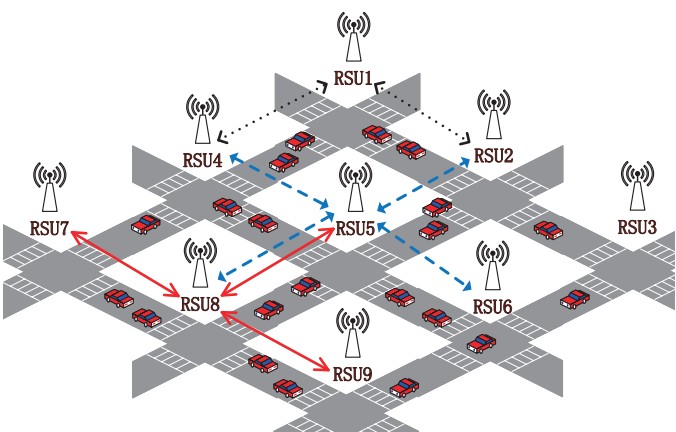

**Figure 4.** Neighbor RSU.

To assess the packet-relaying quality of each street segment between two adjacent RSUs, each RSU at a street intersection needs to periodically send ant packets to its neighbor RSUs, so it is necessary for each RSU to know all its neighboring RSUs. For this purpose, a neighbor RSU discovery protocol is proposed in this paper.

Each RSU periodically broadcasts a neighbor discovery packet NDP, the format of which is shown in the following Figure 5.

| SRC_RSU_ID | DST_RSU_ID | serial_Number | set_of_Street_ID |
|---|---|---|---|
| SRC_CAR_ID | DST_CAR_ID | mode | new_Street_ID |
| data | | | |

**Figure 5.** Packet NDP.

An NDP packet contains the following fields.

(1)　SRC_RSU_ID: ID of the source RSU that issued the NDP packet.
(2)　DST_RSU_ID: ID of the destination RSU that received the NDP packet.
(3)　serial_Number: Every RSU assigns numbers sequentially to the NDP packet it creates.
(4)　set_of_Street_ID: The set of ID of the street on which the neighbor RSUs have been found.
(5)　SRC_CAR_ID: ID of the source vehicle that forwarded the NDP packet in each relay transmission.
(6)　DST_CAR_ID: ID of the destination vehicle that received the NDP packet in each relay transmission.
(7)　mode: The value of mode can be one of the three values of 0, 1 and 2, which, respectively, represent the three transmission modes of NDP data packets in this protocol, as shown in Figure 6: from RSU to vehicle, from vehicle to vehicle, and from vehicle to RSU.
(8)　new_Street_ID: ID of street associated with the newly discovered neighbor RSU.

The tuple values of *RSU_ID* and *serial_Number* fields are used to uniquely identify an NDP broadcast packet, so that the vehicle node can ignore duplicate broadcast packets.

The global configuration parameters of each RSU include $t_{ndp}$ and *st*, which are the time between launching successive NDP and the set of neighbor RSUs, respectively. In the initial configuration phase of the RSU, when the quality of the street segment connectivity is good, a timer may be set to run the neighbor RSU discovery protocol for a period of time to fully discover the neighbor RSUs, and store them in the configuration parameter *st*. RSU NDP broadcasting and receiving procedures are shown in Algorithms 1 and Algorithm 2, respectively.

---

**Algorithm 1** RSU NDP broadcasting.

1: Set timer $T_n \leftarrow t_{ndp}$ ;
2: **repeat**
3:     Generate a NDP broadcast packet *pkt*;
4:     *pkt.SRC_RSU_ID* ← source RSU ID;
5:     *pkt.DST_RSU_ID* ← empty;
6:     *pkt.serial_Number* ←incrementing integer;
7:     *pkt.set_of_Street_ID* ← *st*;
8:     *pkt.SRC_CAR_ID* ← empty;
9:     *pkt.DST_CAR_ID* ← empty;
10:     *pkt.mode* ← 0;
11:     *pkt.new_street_ID* ← empty;
12:     **if** the timer reaches a beacon interval **then**
13:         broadcast the packet *pkt*;
14:     **end if**
15: **until** $T_n == 0$

---

**Algorithm 2** RSU NDP receiving.

1: *rp* ← NDP packet received
2: **if** *rp.DST_RSU_ID* ==this RSU ID **then**
3:     **if** *rp.mode* == 2 **then**
4:         store the *rp.newStree_ID* to the set *st*;
5:     **end if**
6: **end if**

---

In order to avoid repeating the same NDP packet on the same street, after a certain RSU sends out a broadcast packet *pkt* for discovering its RSU neighbors, vehicle nodes that receive this broadcast packet on newly detected streets do not immediately forward the packet, but keep silent for a period of random time before relaying the packet. If an NDP with the same *SRC_RSU_ID* and *serial_Number* value as *pkt* is received again during the silent period, it indicates that the some vehicle node on the same street has forwarded the packet *pkt*, so the vehicle node cancels forwarding each of them, otherwise it forwards *pkt* immediately after the silent period.

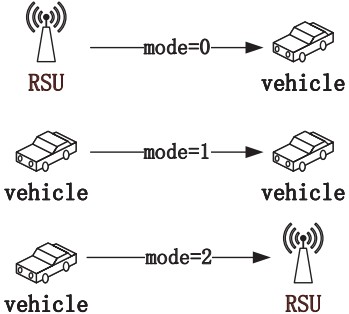

**Figure 6.** Relay mode.

For example, as shown in Figure 7, the dotted circle, the dashed circle, and the half-dashed circle are the signal coverage of *RSU*1, vehicle node *A*, and vehicle node *B*, respectively. *A* and *B* on the street segment between *RSU*1 and *RSU*2 are within the signal coverage of *RSU*1, so they can receive the broadcast packet sent by *RSU*1. Suppose that *A* and *B* receive the packet *rp* sent by *RSU*1, they do not immediately forward it, but each keep silent for a random period of time. Suppose *A* first ends the silent period and forwards *rp*, and *B* receives the *rp* sent by *A* during the silent period and cancels the forwarding of each of them.

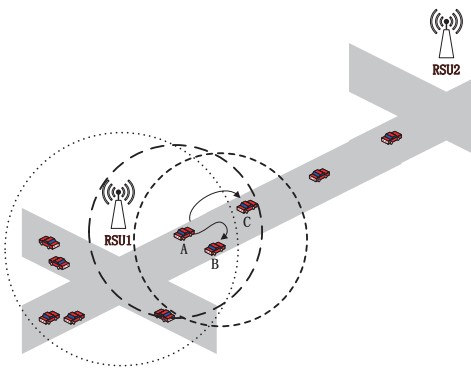

**Figure 7.** Vehicle node forwarding.

The vehicle node forwarding packet will search the vehicle node on the same street, which is the farthest neighbor from the source RSU as the next hop. The operation continues until the packet is transmitted to the neighbor RSU. This procedure is shown in Algorithm 3, where the mode field of the NDP packet plays an important role, as explained from the following two aspects.

---

**Algorithm 3** Vehicle forwarding the received NDP messages.

---

1: *rp* ← NDP packet received
2: *sid* ← this car's street ID;
3: *cid* ← this car's ID;
4: **if** *sid* is not in the set *rp.set_of_Street_ID* **then**
5:    **if** *rp.mode*==0 **then**
6:       wait a random time;
7:       **if** this car receives another NDP packet with the same value for *rp.SRC_RSU_ID* and *rp.serial_Number* **then**
8:          return;
9:       **end if**
10:    **end if**
11: **end if**
12: **if** *rp.mode*==1 **and** *rp.DST_CAR_ID*==*cid* **or** *rp.mode*==0 **then**
13:    **if** a non-source RSU in the neighbor table **then**
14:       *rid* ← the ID of the non-source RSU;
15:       *rp.DST_RSU_ID* ← *rid*;
16:       *rp.SRC_CAR_ID* ← *cid*;
17:       *rp.DST_CAR_ID* ← empty;
18:       *rp.newStree_ID* ← *sid*;
19:       *rp.mode* ← 2
20:    **else**
21:       **if** a car *dc* that is in the neighbor table of this car, on the same street as this car, further away from the source RSU than this car, and farthest from the source RSU **then**
22:          *rp.DST_RSU_ID* ← empty;
23:          *rp.DST_CAR_ID* ← the ID of *dc*;
24:          *rp.SRC_CAR_ID* ← *cid*;
25:          *rp.newStree_ID* ← *sid*;
26:          *rp.mode* ← 1
27:       **else**
28:          store *rp* to message buffer queue of this car, and check the buffer queue and resend *rp* periodically
29:       **end if**
30:    **end if**
31: **end if**

---

Firstly, when the source RSU starts to broadcast the NDP packet, the value of the mode field is set to 0. In this mode, the NDP packet will be broadcast from the RSU node to the nearby vehicle nodes, and the NDP packet has no determined forwarding target vehicle. There may be several vehicles receiving the NDP packets on the same street segment, and only one of them needs to forward it on the same street segment. Therefore, vehicle nodes will use a backoff algorithm to avoid repeating the forwarding of NDP packets on the same street segment. Assuming that the first vehicle selected for forwarding the NDP packet on a certain street segment is called *SC*, if there is a non-source RSU in the neighbor table of *SC*, it means that the distance between the two RSUs is very close, and the neighbor RSU can be reached by forwarding packets through only one vehicle (of course, it is very rare for these RSUs to be so close in actual deployment). If there is a non-source RSU in the neighbor table of the vehicle, it means that the neighbor RSU is found, the value of mode needs to be modified to 2, and then the NDP packet is sent from the vehicle to the destination RSU, and other vehicles will ignore the NDP packet with the value of mode of 2.

Second, when the value of mode is 1, it means that the NDP packet will be forwarded from vehicle to vehicle. The NDP packet will contain the ID of the next hop vehicle, so the backoff algorithm is no longer needed. In this mode, if the ID of the vehicle receiving the NDP packet is the same as *DST_CAR_ID* in the NDP packet, the vehicle will accept the NDP and then try to forward it, otherwise it will ignore the packet. If the vehicle node receiving the NDP packet finds a non-source RSU in its neighbor table, it means that a neighbor RSU is found, so the mode in the NDP packet is set to 2, indicating that the NDP packet will be forwarded from the vehicle to the RSU node. If the vehicle node does not find the non-source RSU in its neighbor table, the vehicle node farthest from the source RSU on the same street segment needs to be searched in its neighbor table as the next hop vehicle node, and the mode value of this vehicle node is still 1. Through this NDP forwarding method, when approaching the destination RSU, the sender vehicle of the last hop to the destination RSU is not necessarily the closest to the destination RSU, but the destination RSU must be within the signal coverage of the sender vehicle, and the NDP packet successfully reaches the destination RSU.

In a word, according to the mode field in the NDP packet, Algorithm 3 can perform fine-grained control on NDP packet forwarding to ensure that the packet can be forwarded to the neighboring RSU node in the presence of an appropriate traffic flow.

### 2.3. Calculation of Ant Pheromone

The adjacent RSUs at both ends of a street segment send ant packets to each other periodically. The ant packets can reach the RSU at the other end through the greedy forwarding between vehicle nodes, and the pheromone value is calculated according to the delay of the ant packet. We optimize the update of the ant pheromone [6] by adding a decay function over time and arithmetic mean. The pheromone value of the street segment from intersection *i* to intersection *j* will be updated using the following formula:

$$phero_{ij} = \frac{phero_{ij} + \Delta phero_{ij}}{1 + \Delta phero_{ij}} \tag{1}$$

where $\Delta phero$ is

$$\Delta phero_{ij} = C + \frac{2}{\pi} \arctan(\frac{minDelay_{ij}}{delay_{ij}}) \tag{2}$$

where *C* is a constant less than 0.5. $delay_{ij}$ and $minDelay_{ij}$ are the time and minimum time it takes the ant packet to traverse the street segment from intersection *i* to intersection *j*, respectively.

To describe the reduction of the ant pheromone over time, i.e., pheromone evaporation, a decay function over time is defined as follows:

$$decay(t) = \frac{1}{1 + \alpha \times t} \tag{3}$$

where $\alpha$ is the time decay constant ant $t$ is the delay between RSU receiving two ant packets. Each RSU decreases the ant pheromone of related street segments using the following formula:

$$phero_{ij} = phero_{ij} \times decay(time_{ij}) \tag{4}$$

The RSU is located near each road intersection, so it receives ant packets from different street segments, and calculates the pheromone values for different street segments. Then, the RSU needs to send the pheromone value of each street segment to the SDN server, and for this purpose, the RSU constructs a corresponding message $mp$ for each street segment, which encapsulates the value of the pheromone, the generation time, and the two-tuple consisting of the sender RSU and receiver RSU. After that, the RSU sends $mp$ to the SDN server.

The SDN server sets a $mp$ message buffer for each street segment, which can buffer at most $M$ messages closest to the current time. If the message is not eliminated in the message buffer for a time interval much larger than the transmission period of the ant packet, it indicates that the connectivity quality of the street segment is poor. The poor case can also cause the number of messages in the message buffer to be less than $M$. The connectivity quality of the street segment from $RSU_i$ to $RSU_j$ is estimated as follows:

$$cq_{ij} = \frac{1}{ave\_phero_{ij} \times decay(ave\_time_{ij})} \tag{5}$$

where $ave\_phero_{ij}$ and $ave\_time_{ij}$ are the arithmetic average of the ant pheromone values and arithmetic average of the residence time of all messages in the message buffer corresponding to the street segment from intersection $i$ to intersection $j$, respectively.

Assuming that the number of RSUs in the simulation area is $N$, the evaluation values of the connectivity quality of all street segments can be stored in a $N \times N$ matrix $A$ as follows:

$$A = \begin{bmatrix} \infty & cq_{12} & \cdots & cq_{1n} \\ \mathbf{cq_{21}} & \infty & \cdots & cq_{2n} \\ \vdots & \vdots & \ddots & \vdots \\ \mathbf{cq_{n1}} & \mathbf{cq_{n2}} & \cdots & \infty \end{bmatrix} \tag{6}$$

The initial value of each element in $A$ is infinite, denoted as $\infty$. Since the adjacent RSUs send ant packets to each other, the two adjacent RSUs independently evaluate the connectivity quality of the same street segment according to the received ant packets from opposite directions, which may lead to inconsistent evaluation values, i.e., $cq_{ij} \neq cq_{ji}$.

In the routing scheme proposed in this paper, the evaluation value of each street segment is unique. Therefore, to be conservative, the minimum value of the evaluation values in different directions of each street segment are taken as the evaluation value of this street segment. In (6), assuming that the bolded matrix element value $cq_{ij}$ is smaller than $cq_{ji}$, the matrix A is converted into $A'$ as shown below:

$$A' = \begin{bmatrix} \infty & \mathbf{cq_{21}} & \cdots & \mathbf{cq_{n1}} \\ \mathbf{cq_{21}} & \infty & \cdots & \mathbf{cq_{n2}} \\ \vdots & \vdots & \ddots & \vdots \\ \mathbf{cq_{n1}} & \mathbf{cq_{n2}} & \cdots & \infty \end{bmatrix} \tag{7}$$

### 2.4. SDN-Based Routing

When the source vehicle $sv$ needs to send a data packet to the target vehicle $dv$, the next hop of the packet must be found. For this purpose, it is necessary to determine whether the $sv$ and the $dv$ are on the same street segment. In the case of the same street segment, if the $dv$ is just in the neighbor table of the $sv$, the next hop is the $dv$, otherwise, the next hop is calculated by the greedy forwarding algorithm, i.e., the next hop is the node that is closest to the $dv$ on the same street segment in the neighbor table of the $sv$. Additionally, in the case where the $sv$ and the $dv$ are not on the same street segment, the next hop of the packet is calculated according to the shortest anchor path.

The shortest anchor path is an RSU sequence calculated by the SDN server according to the starting RSU and the ending RSU. The first RSU through which the $sv$ forwards the packet is called the starting RSU, and the last RSU through which the packet arrives at the $dv$ is named the ending RSU. The starting RSU and the ending RSU are the first and last elements of the shortest anchor path, respectively.

The starting RSU $sr$ and ending RSU $dr$ are elected based on their distance to the source vehicle $sv$ and the target vehicle $dv$. First, the RSUs at both ends of the street segment where the $sv$ is located are selected as candidate starting RSUs, of which the RSU closer to the $dv$ is elected as the starting RSU. Similarly, the RSUs at both ends of the street where the $dv$ is located are determined as the candidate ending RSUs, of which the RSU closer to the starting RSU is selected as the ending RSU.

After the starting RSU $sr$ and ending RSU $dr$ are determined, the source vehicle node $sv$ constructs a routing request (RREQ) packet with the control fields $sr$ and $dr$, then sends the RREQ to the SDN server, which has global information of the streets' connectivity. Then, SDN server calculates the shortest anchor path $sp$ from $sr$ to $dr$ by Dijkstra's algorithm based on (7). The $sp$ consists of a sequence of RSU from the starting RSU to the ending RSU. The SDN encapsulates the $sp$ in the routing reply (RREP) packet and returns it to the source vehicle $sv$. After receiving the RREP, the source vehicle $sv$ reads the $sp$ in the RREP, encapsulates it in the header of the packet, and sent it to target vehicle $dv$. The packet will be forwarded along the $sp$ until the target vehicle $dv$. When transmitting data packets between two adjacent RSUs, the greedy forwarding algorithm between vehicles is used. The vehicle to vehicle/RSU packet forwarding strategy is shown in Algorithm 4, and the RSU to vehicle packet forwarding strategy is shown in Algorithm 5.

---

**Algorithm 4** Vehicle to vehicle/RSU packet forwarding.

---

1:   $nextHop \leftarrow$ empty;
2:   **if** $sv$ and $dv$ are on the same street **then**
3:     **if** $dv$ is in the neighbor table of $sv$ **then**
4:       $nextHop \leftarrow dv$;
5:     **else**
6:       choose $nextHop$ with the greedy forwarding to $dv$;
7:     **end if**
8:   **else**
9:     $nextRSU \leftarrow$ get next RSU from the $sp$;
10:    **if** $nextRSU$ is in the neighbor table of $sv$ **then**
11:      $nextHop \leftarrow nextRSU$;
12:    **else**
13:      choose $nextHop$ with the greedy forwarding to $nextRSU$;
14:    **end if**
15:   **end if**
16:  **if** $nextHop$==empty **then**
17:    store the packet to buffer queue of this node;
18:  **end if**

---

---

**Algorithm 5** RSU to vehicle packet forwarding.

---

1:  *nextHop*← empty;
2:  **if** this RSU is the ending RSU  **then**
3:      **if** *dv* is in the neighbor table of this RSU **then**
4:          *nextHop*← *dv*;
5:      **else**
6:          choose *nextHop* with the greedy forwarding to *dv*;
7:      **end if**
8:  **else**
9:      *nextRSU*← get next RSU from this RSU;
10:      *streetID*← ID of street between this RSU and *nextRSU*;
11:      *nextHop*← the node closest to the *nextRSU* on the street *streetID*;
12:  **end if**
13:  **if** *nextHop*==empty **then**
14:      store the packet to buffer queue of this RSU;
15:  **end if**

---

## 3. Results and Discussion

We carried out extensive simulations in Veins (Vehicles in Network Simulations), which is an open source simulation platform for running vehicular network simulations. Veins integrates two well-established simulators: OMNeT++ and SUMO, where OMNeT++ is an object-oriented event-based simulation library and framework for building network simulators, and SUMO is a microscopic traffic simulation package. Veins provides the connection to SUMO by Traffic Control Interface (TraCI).

The simulation scenario is a grid area covering 3600 m × 3000 m, where the distance between two adjacent intersection is set to 600 m. There are 38 street segments and 24 intersections in total. Approximately 50 meters away from each intersection, there exits an RSU. Each street segment is two-way, with one lane in each direction. The simulation time is 1200 s, and the warm-up period is set to 160 s. The simulation parameters are summarized in Table 1.

**Table 1.** Simulation parameters.

| Parameter | Value |
| :---: | :---: |
| Simulation area | 3600 m × 3000 m |
| Simulation time | 1200 s |
| Warm-up period | 160 s |
| Mac protocol | IEEE 802.11p |
| Number of intersections | 24 |
| Number of street segments | 38 |
| Signal transmission radius | 260∼ 320m |
| Packet sending rate | 1∼12 pkt/s |
| Maximum vehicle speed | 6∼22 m/s |
| Time decay constant $\alpha$ | 0.0001 |

In order to evaluate the performance of the routing protocols, we focus on two metric properties: the packet delivery ratio (PDR) and average end-to-end delay (EED), where PDR is calculated as following: PDR =number of packets received/total number of packets sent, and EED is defined as the average delay of packets transmission from the source node to destination node. These two metric properties of our proposed scheme are evaluated within the confidence interval of 95%. The confidence interval is computed out of 10 simulation runs for each different random number seed. We compare the proposed scheme with GPSR and GSR, and analyze the simulation results.

Figure 8 shows the general upward trend of the packet delivery ratio as the signal transmission radius increases. In this simulation scenario, the maximum vehicle speed is set to 10 m/s. Due to the change of vehicle communication range, more nodes can be

selected to relay data packets, which affects the transmission of ant packets to a certain extent, and then affects the ant pheromone value of the street segment, thus dynamically affecting the routing of data packets, so the packet delivery ratio will fluctuate slightly.

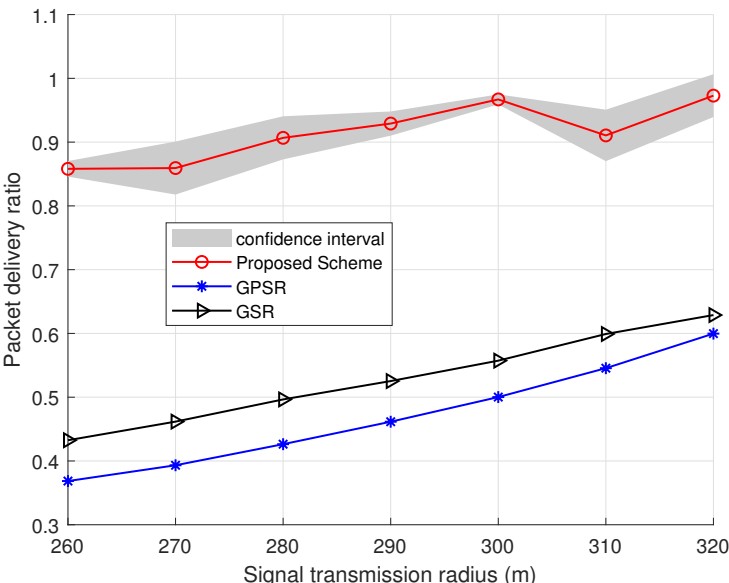

**Figure 8.** Packet delivery ratio with the signal transmission radius. Confidence interval with $\alpha = 0.05$.

Figure 9 presents the variation of the packet delivery ratio with the vehicle speed. With the increase of vehicle speed, the packet delivery ratio fluctuates to some extent, which indicates that the vehicle speed affects the transmission of data packets. The change of vehicle speed will affect the distribution of vehicle neighbor nodes at a certain time, thus affecting the selection of the next relay node of the packet, and then affecting the packet delivery ratio.

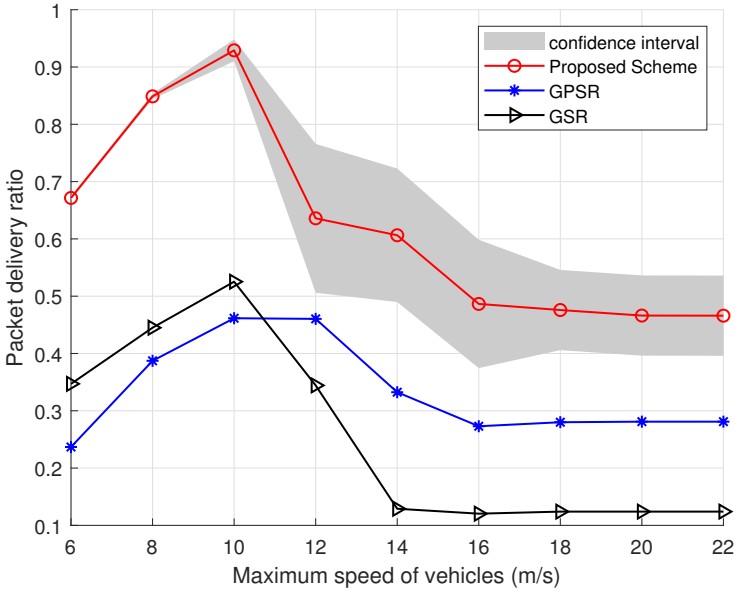

**Figure 9.** Packet delivery ratio with the maximum speed. Confidence interval with $\alpha = 0.05$.

Figure 10 depicts the packet delivery ratio for different packet rates. When the packet rate increases, that is, the number of data packets sent per second increases, the packet delivery ratio may increase in the initial stage, but when the packet rate keeps increasing,

the network communication is heavily loaded, which will lead to no increase or decrease in the packet delivery ratio.

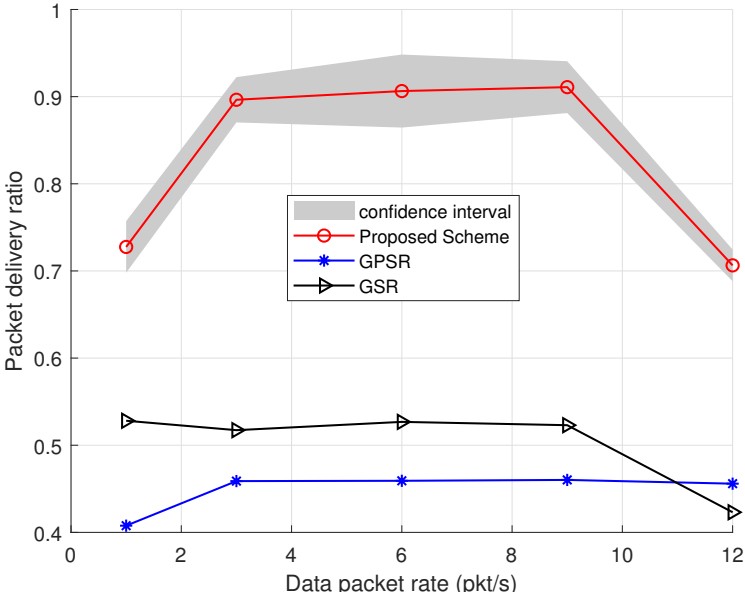

**Figure 10.** Packet delivery ratio with the data packet rate. Confidence interval with $\alpha$ = 0.05.

Figure 11 shows the average end-to-end delay against the signal transmission radius. In this simulation scenario, the maximum vehicle velocity is set to 10 m/s. The proposed routing scheme always selects the street segment with low delay, so it results in stable low delay. However, the delay time of the scheme presented in the Figure 11 is the delay time for the source vehicle node to send packets to the destination vehicle after obtaining the anchor path, excluding the route request time, the time for the SDN server to calculate the anchor path and the route response time. If the above time is included, the delay time is greater than that of the other two routing schemes.

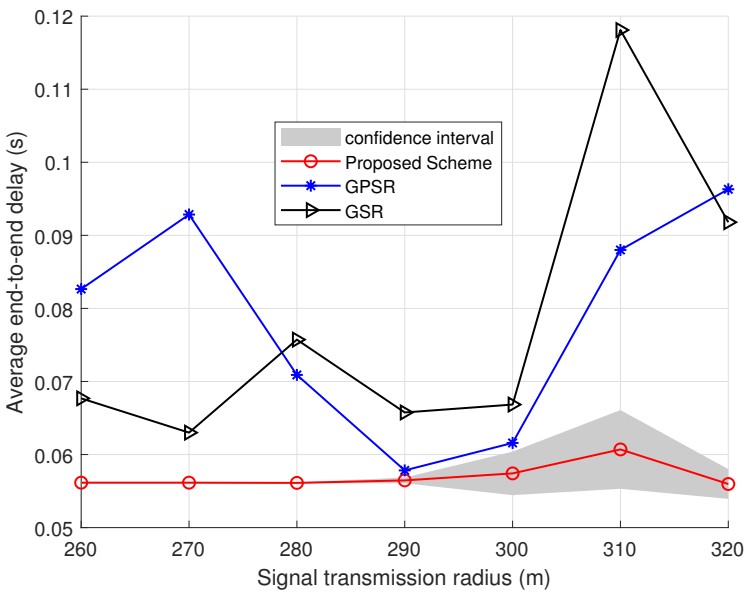

**Figure 11.** Average delay with the signal transmission radius. Confidence interval with $\alpha$ = 0.05.

The IEEE 802.11p protocol uses the carrier sense multiple access/collision avoidance (CSMA/CA) mechanism for wireless terminal media access. In the dense traffic scenario, when a large number of nodes compete for channel resources at the same time, the conflict

will increase, and the transmission performance of the network will decline. In Figure 11, the average end-to-end delay of the proposed scheme fluctuates when the signal transmission radius expands to about 310 m, which corresponds to the fluctuation of the packet delivery ratio of the proposed scheme when the signal transmission radius is also about 310 m in Figure 8. The possible reason is that under the current simulation configuration, the vehicle density in the signal coverage area increases when the transmission radius expands to about 310 m. A large number of vehicle nodes compete for channel resources, which causes the fluctuation of the network performance.

## 4. Conclusions

A intersection-based unicast routing protocol using ACO in SDN was proposed. In the proposed routing scheme, a single-hop RSU neighbor discovery protocol was proposed, through which each RSU can obtain all its neighboring RSUs. Then, using ant packets, RSUs regularly evaluated the connectivity of the street segments between adjacent RSUs, and sent the evaluation information to the SDN server, which has the global information of each street segment in the simulation area. According to this evaluation information, the SDN server can calculate the optimal anchor path from the sending vehicle node to the destination vehicle node, and feed it back to the sender. Based on this path, the sender forwards the packet to the destination vehicle. Experimental results show that the proposed scheme can effectively improve the packet delivery rate compared with other typical routing schemes. In future work, we will focus on the security of routing to prevent malicious attacks from hackers.

**Author Contributions:** Conceptualization, G.Z. and H.Z.; methodology, J.L.; software, H.Z.; validation, J.L., L.J. and H.Z.; formal analysis, L.J.; investigation, J.L.; resources, L.J.; data curation, H.Z.; writing—original draft preparation, H.Z.; writing—review and editing, H.Z.; visualization, J.L.; supervision, G.Z.; project administration, G.Z.; funding acquisition, G.Z. All authors have read and agreed to the published version of the manuscript.

**Funding:** This research was funded by the National Natural Science Foundation of China (No. 61971245) and Doctoral Research Foundation of Nantong University (No. 135422630029).

**Data Availability Statement:** Data sharing is not applicable to this article.

**Conflicts of Interest:** The authors declare no conflict of interest.

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
