# Peer review of "Intersection-Based Unicast Routing Using Ant Colony Optimization in Software-Defined Vehicular Networks"

_electronics, doi:10.3390/electronics12071620_

Round 1

Reviewer 1 Report

The article proposes an SDN based unicast routing scheme for VANET in urban traffic environment to address local optimum and network congestion problems in traditional geographic routing protocols. The scheme uses Dijkstra's algorithm to find a global optimal anchor path and RSU neighbor discovery protocol to discover and evaluate the communication connection quality of each street segment. The proposed solution uses Ant Colony Optimization and is based on a single-hop RSU neighbor discovery protocol. It improves packet delivery rate compared to other typical routing schemes.

In my view, the approach proposed in the article is both correct and useful, as evidenced by the significant achievements demonstrated in the results. However, it is worth noting that the investigation was limited to a single simulation environment, which could be viewed as a potential weakness. It may be beneficial to conduct additional analyses and document their findings. Despite this, I believe that the article is of good quality overall and recommend its acceptance after making some minor corrections.

There are some typos, e.g. “procdure” in line 166 and line 186, and “standlone” in line 115.

Figure 8 is way too crowded.

Author Response

Thank you for your constructive and professional comments that helped me improve my paper. The modification is in the attachment. Thank you.

Reviewer 2 Report

In line 115 standlone should be standalone

I think that is interesting, but there are several aspects that are not clear in the text.

It is a bit confusing how the RSU can communicate. In the text, the authors commented that hop by hop using the cars. Is this true? If the RSU have 5G connection, it has sense to use this network to communicate the different RSUs.

The discovery of the neighbor RSUs can be done using the 4G/5G network, I can understand the necessity of to send periodic packets to measure the link quality, but the discovering can be done using the information of the SDN server.

In the paragraph between the lines 183-186. How the vehicle know the next hop node that is close to the destination RSU?

In section 2.4

In any routing protocol, a very important aspect is how the information about the nodes is collected and how the routing tables of the nodes are actualized, and this information is not in the text.

In this section, the authors connected that is the SDN sever that computes the route and inform the route to the nodes, but how the information about the nodes is transmitted to the SDN sever or how the SDN server inform to the different nodes that are part of the route about the next hop is not included, and this is a crucial part of any routing protocol. Information about this must be included in the text. 

How the sever know the position of the nodes? What is the periodicity of actualization of the positions? How the routing information is communicated to the nodes that are part of the network?

This information also helps to understand the overload that the protocol will introduce in the network.

In section 3

How many simulations with different seeds have been done?

In general, it is necessary to include the figures the confidence interval.

In figure 10 there is a very strange behavior, the PDR raise, later fall and finally raise. There isn’t a tendency. Do have the authors some theory about this strange behavior?

In the figure 12, due to the small differences in of the different protocol, it is necessary to include the confidence interval.

Author Response

(The authors gave the same response as above.)

Round 2

Reviewer 2 Report

I found this a bit confusing, “A neighbor RSU is defined as a single-hop RSU with only one street segment located
 between them”. In a wireless network, one-hop neighbors are the nodes that can communicate between them without the necessity to relay the packets by intermediate nodes. It can be assumed that neighbors RSU can communicate between them. I have assumed that the concept of neighbor is two RSUs that are in the same street segment and doesn’t exist any additional RSU in the middle.

“RSU4,RSU6” is necessary a space.

In Algorithm 2, if the destination in the NDP is set to empty
This condition can be false if the vehicle that sends the packet has not already actualized the neighbor RSU table.
if rp.DST_RSU_ID ==this RSU ID then

Who do the nodes actualize the neighbor RSU table? I suppose that exists a life timeout parameter.

There is a thing that I don’t understand, and it is how the position of the cars is known by the controller. How does the controller know on which street is the car?

Figure 12 has a very strange behavior for some protocols, have the authors some hypotheses about this behavior?

Author Response

Thank you for your constructive and professional comments that helped me improve my paper. Thanks for your hard working.
